# Female community health volunteers' knowledge and confidence in providing community-based diabetes self-management support in Nepal: A biphasic mixed method evaluation

Ada Thapa[1,2], Susagya Bhusal[1*], Omkar Jha[1], Adipti Pantha[1], Sudim Sharma[1], Austin Seals[3], Madhav K. C.[4], Archana Shrestha[5], Bibhav Acharya[6], Yashashwi Pokharel[1,3,7]

1 Health Foundation Nepal, Dang, Nepal, 2 Center for Integration Science in Global Health Equity, Department of Medicine, Brigham and Women's Hospital, Boston, Massachusetts, United States of America, 3 Departments of Internal and Cardiovascular Medicine, Atrium Health Wake Forest Baptist, Wake Forest University, School of Medicine, Winston-Salem, North Carolina United States of America, 4 Department of Epidemiology, University of Nebraska Medical Center, Omaha, Nebraska, United States of America, 5 Department of Public Health, Kathmandu University School of Medical Sciences, Dhulikhel, Nepal, 6 University of California, San Francisco, California, United States of America, 7 Department of Implementation Science, Office of Global Health, Atrium Health Wake Forest Baptist, Wake Forest University, School of Medicine, Winston-Salem, North Carolina, United States of America

☯ These authors contributed equally to this work.
* bhusal.susagya06@gmail.com

## Abstract

Self-management support by community health workers, referred to as Female Community Health Volunteers (FCHVs) in Nepal, can enhance diabetes care in resource-constrained settings. We assessed FCHVs' knowledge, perceived self-efficacy, and barriers to supporting diabetes self-management before and after training. Guided by the Health Belief Model, we conducted a biphasic, Qual+quan, concurrent, embedded mixed-methods evaluations among 28 FCHVs in four wards in rural mid-Western Nepal. We evaluated the program into two phases, each consisting of focus group discussions (FGDs); trainings that included didactics, hands-on-practice, and role play demonstrations; and pre- and post-training surveys. Phase 2 occurred nine months after Phase 1. We conducted trainings and evaluations over 4–5 hours/ward/phase. FGD data were analyzed using both deductive and inductive coding and emerging themes were then examined for patterns and relationships among the codes. We summarized quantitative data using descriptive statistics and integrated with qualitative data during interpretation. FCHVs' mean age was 48 years, 32.1% had completed secondary education, and 75% had served as FCHVs for >10 years. We identified five themes in qualitative analysis: (1) diabetes knowledge, (2) diet and lifestyle counseling, (3) FCHV's confidence, (4) training and education, and (5) barriers and motivators for community-based self-management support. Mixed-methods evaluations showed knowledge of diabetes and confidence in self-management skills

**Data availability statement:** All relevant data are within the paper and its Supporting Information files.

**Funding:** This work was supported by the Health Foundation Nepal to YP. YP is also supported by the National Institutes of Health Awards K23HL171954, 1R01HL173523, and by the Patient-Centered Outcome Research Institute Award BPS-2023C1-31377. BA is supported by the National Institutes of Health Awards R01MH133231, R01AA029303, R01MH135376, and R01TW012682. The funders had no role in study design, data collection and analysis, decision to publish, or preparation of the manuscript.

**Competing interests:** Dr. Bibhav Acharya is an academic editor for PLOS GPH. The other authors declare no competing interests. This does not alter our adherence to PLOS ONE policies on sharing data and materials.

improved after the training. Although barriers, such as inadequate incentives, persisted in Phase 2, communities' trust in FCHV as diabetes self-management supporter improved. These findings suggest that FCHVs may be willing to take a greater role in diabetes self-management support, and that even short training sessions have the potential to enhance their confidence and knowledge, particularly if our findings are further substantiated. Ensuring appropriate incentives, ongoing training and system-level support are important for sustainability of such programs.

## Introduction

Non communicable diseases (NCDs) such as heart disease, stroke, and diabetes are the leading causes of mortality and morbidity globally [1]. Approximately 106.9 million adults aged 20–79 years are living with diabetes in South-East Asia. This number is projected to increase by 73% reaching 185 million by 2050 [2]. Nepal, a small developing country in South- Asia, with a population of 26 million has the fourth highest number of adults aged 20–79 years with diabetes [3]. The number of adults with diabetes in Nepal is predicted to increase from 1.3 million to 2.4 million by 2050 [3,4]. Although the burden is increasing rapidly, there are limited resources and healthcare capacities to meet the growing demand for diabetes care. Managing diabetes in these settings presents numerous challenges, including limited healthcare professionals, medications, and challenges in improving patients' self-management skills. Given these challenges, community-based, sustainable and scalable diabetes care approaches that do not solely rely on traditional healthcare providers can augment diabetes care [5,6].

Improving diabetes self-management skills within communities is an important but neglected aspect of diabetes care. Self-management skills enhance knowledge about diabetes, medication adherence, adoption of healthy lifestyle modifications and confidence in chronic disease management [7,8]. Community-based diabetes self-management support by community health workers (CHWs) therefore is highly relevant in low-resource settings [5,6]. CHWs serve as a vital link between healthcare systems and communities, facilitating services and providing culturally appropriate care [9]. By leveraging local knowledge and trust, CHW programs effectively provide contextually appropriate services within communities. Nevertheless, aligning the fitness of CHW programs to local context, including CHW's needs, is important, particularly when CHWs are involved in new areas of healthcare like diabetes. Female Community Health Volunteers (FCHVs) are the most common types of CHW in Nepal and have been an integral part of the healthcare system since 1988. There are ~ 50,000 FCHVs in Nepal chosen by local women-led civic group called the Mother's group [10,11]. FCHVs typically receive basic training from the Ministry of Health and Population focusing on promoting healthy behaviors among communities, safe motherhood, child health, family planning, and distributing contraceptives, oral rehydration solutions packets, vitamin A capsules, iron tablets, treating pneumonia, referring severe cases to health institutions, and providing motivation and education

[5,10–13]. Recently there has been growing interest in expanding their involvement to non-communicable diseases (NCDs), including diabetes self-management support [9,14–17]. However, before expanding such programs, evaluating whether FCHVs possess the necessary knowledge and confidence in delivering diabetes care is essential. An assessment of their training needs, potential barriers, and ability to retain information is important to understand the program's success. This is particularly relevant given the considerable variability in FCHV's education and background [14,16]. Therefore, we conducted a biphasic concurrent, embedded mixed-methods evaluation to understand FCHVs' knowledge, perceived self-efficacy and barriers in providing diabetes self-management support and to evaluate the impact of a short training on change in knowledge at various time points.

## Materials and methods

### Ethics statement

We obtained ethical approval from Nepal Health Research Council (Ref: 2483) and a written informed consent from each FCHV.

### Setting and sampling

Health Foundation Nepal (HFN), a non-profit organization, has been carrying out a community-based prospective epidemiological and implementation program, NCD Nepal study, in four wards of Dang District, located Southwest of Kathmandu, the capital [18]. The NCD Nepal study is an ongoing longitudinal study where FCHVs project is embedded. The NCD Nepal study collaborates with FCHVs to coordinate program activities like connecting with the study participants for follow up [18]. Prior to this study, the FCHVs were not trained on diabetes self-management.

We obtained contact information and called all 45 FCHVs from the four wards of the NCD Nepal study area informing about the current study. Due to COVID-19, gathering was restricted to seven people according to local rules. We could not engage FCHVs who were unreachable by phone. Three FCHVs declined to participate due to other commitments. We conducted a second round of calls with FCHVs who had initially agreed to participate. Once seven FCHVs from each ward confirmed participation, we did not contact the remaining FCHVs to comply with the gathering policy. We invited FCHVs to local schools and health centers for training and evaluated their readiness in using community-based self-management support for diabetes, hypertension, hyperlipidemia and tobacco use. The current manuscript focuses only on diabetes. All 28 FCHVs completed the study. In accordance with local norms, we provided each FCHV ~3 USD per phase and a light snack and tote bag containing educational pamphlets about chronic diseases.

### Data collection

We collected both qualitative and quantitative data simultaneously but independently across two phases (Fig 1, Fig 2). Phase 1 took place from January 5, 2021, to January 8, 2021, and phase 2 took place from October 9, 2021, to October 12, 2021. Each phase included a Focus Group Discussion (FGD), a training session, and two diabetes-focused survey questions before and after each training in every ward (Fig 1). We completed all activities in each phase within 4–5 hours.

Incorporating constructs from the Health Belief Model (HBM), specifically knowledge, self-efficacy, and barriers, we designed interview guides and surveys to evaluate FCHVs' understanding of diabetes, their confidence in counseling, and perceived obstacles to providing community-based diabetes self-management support [19,20]. The HBM proposes six socio-behavioral constructs influencing health behaviors. We selected three constructs – knowledge, self-efficacy, and barriers - to understand how FCHVs perceive and provide self-management support reflecting internal and external factors that influence their practices. The survey had five items for knowledge, three items for self-efficacy and one item for barriers. For each phase, we included only relevant items. Our working knowledge of FCHVs informed us in selecting

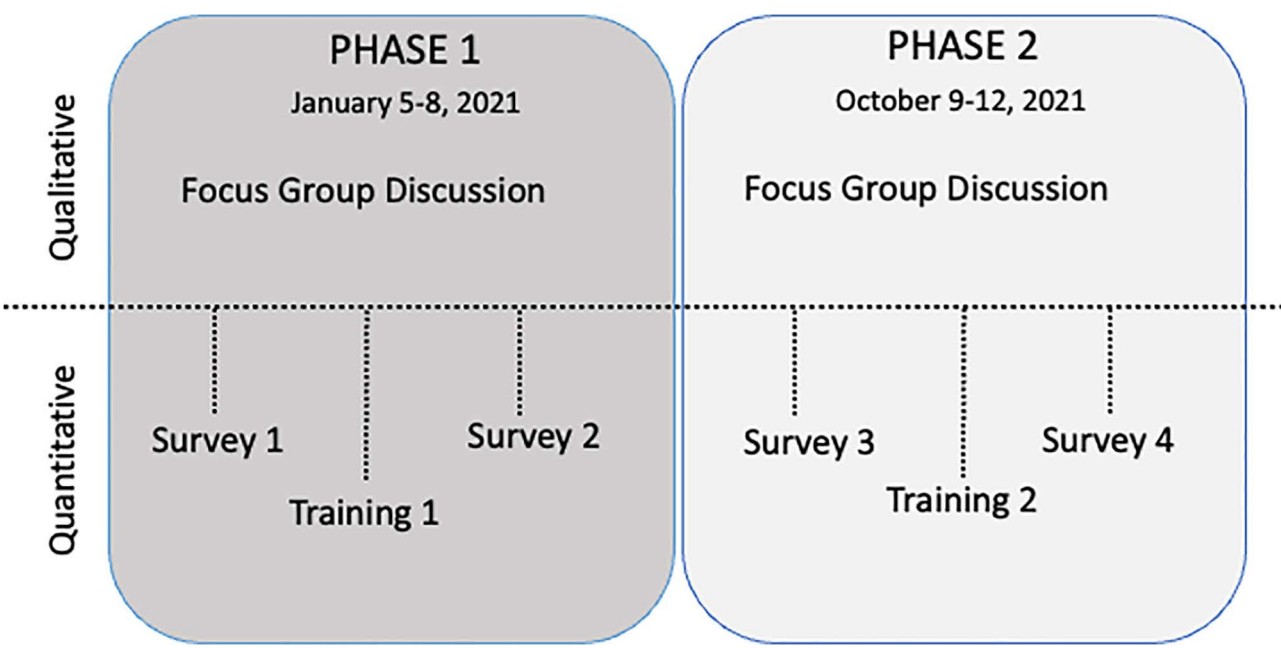

**Fig 1. Study Timeline.**

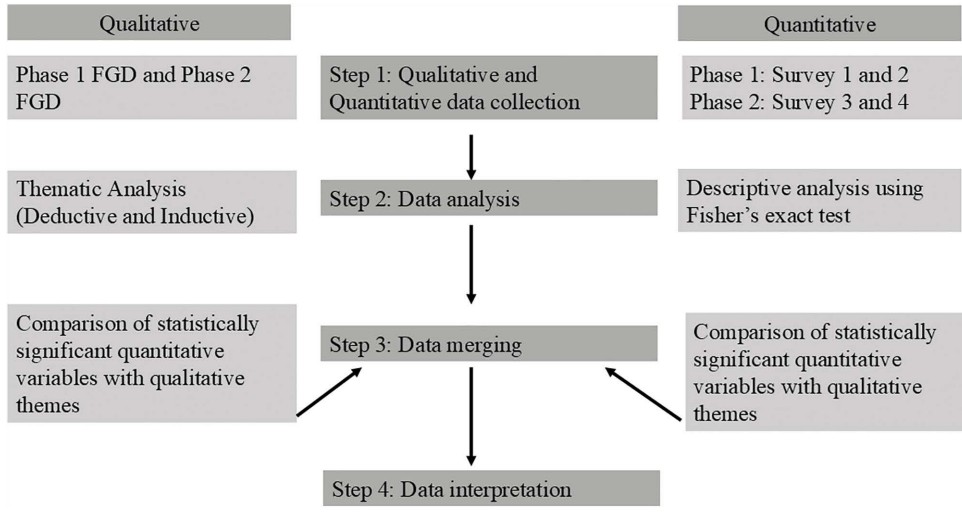

**Fig 2. Flowchart of Mixed Method Design.**

relevant survey questions and responses. We first created the survey in English. SB and SS independently translated them into Nepali language, that was reviewed and reconciled by the larger investigative team. Subsequently, we pilot tested the survey among study staff and refined for clarity and relevance. Then we pre-tested the survey among a few lay individuals with similar health literacy and educational backgrounds as the FCHVs from the study area. This resulted in minor additional refinement for clarity and flow. We used similar process for the interview guides.

### Phase 1

**Focus group discussion.**  We conducted the FGDs in the Nepali language (S2 Text) within local schools and health centers maintaining privacy. We audio-recorded the interviews that lasted between 45–90 minutes.

**Survey 1.**  At baseline, we collected sociodemographic information and administered the 9-item survey (S1 Text) about diabetes concurrently with FGDs.

**Training 1.**  The primary objective of the training was to enhance FCHVs' knowledge and skills, empowering them to provide effective diabetes self-management support. The training program included diabetes-focused didactics, hands-on training exercises (such as glucose measurement), and role-play demonstrations. During the hands-on training, participants were educated on normal, borderline, and high glucose levels. A study physician delivered didactics on diabetes management, while project staff facilitated role-play scenarios simulating physician-patient interactions for diabetes management. The training emphasized diabetes risk factors, sign and symptoms, complications and prevention and control measures, importance and approaches for control, including medications and lifestyle modifications, medication adherence, importance of clinical follow up and community-based care approaches. We used educational materials such as presentations, posters, and pamphlets throughout the session.

**Survey 2.**  Following the training, we administered a second survey (S1 Text), similar to survey 1, consisting of six items to evaluate immediate changes in FCHVs' knowledge after the training. Items no longer relevant at survey 2 were excluded.

### Phase 2

**Focus group discussion.**  Nine months after phase 1, we started phase 2 with FGD (S2 Text) like phase 1. We further evaluated barriers and facilitators that FCHVs encountered while providing diabetes-related care in the community.

**Survey 3.**  We administered the survey (S1 Text) comprised of nine items about diabetes concurrently with FGD.

**Training.**  We conducted a refresher training replica of phase 1 that included diabetes-focused didactics, hands-on training exercises (such as checking glucose levels), and role-play demonstrations.

**Survey 4.**  We conducted survey 4 (S1 Text) following the refresher training that was similar to survey 2.

### Data analysis

**Quantitative analysis.**  We used R version 4.0.2 for descriptive statistics and Fisher's exact test to detect differences across the phases. A two-sided p-value $<0.05$ was considered statistically significant. We did not adjust for multiple comparisons. To evaluate a relatively long-term changes in readiness for diabetes self-management support from our training, we compared the results from surveys 1–4.

**Qualitative analysis.**  We applied both deductive and inductive coding techniques for qualitative analysis of interview data [21]. Deductive codes were based on pre-established constructs from the HBM, while inductive coding facilitated the emergence of new themes from the data. We derived the themes by identifying and interpreting key patterns and relationships among the codes.

AT and SB reviewed the audio-recordings and transcribed interviews to ensure quality and developed a codebook that was shared with the larger investigative team for finalization. AT then independently applied the finalized codes using NVivo 1221. We developed code summaries to identify emergent themes, which were consolidated by both AT and SB. These themes were then discussed with the broader team to ensure consensus and further refinement of the findings. We aligned the emerging themes along the three HBM constructs.

**Mixed-methods integration.**  We employed concurrent mixed-method approach, embedding quantitative data with qualitative dominant data (Qual + quan), collecting and analyzing qualitative and quantitative data separately but simultaneously in two phases [22]. We integrated the data together during results interpretation for convergence and complementarity (Fig 2) [22].

## Results

### Sociodemographic characteristics

FCHVs' mean age (SD) was 48 (6.8) years. One FCHV was not able to read and write (Table 1).

### Main results

We compiled qualitative and quantitative results under five themes: diabetes knowledge; diet and lifestyle counseling; FCHVs' confidence; training and education; and barriers and motivators in community-based counseling delivery (Table 2).

### Theme 1: Diabetes knowledge

FCHVs resoundingly expressed having a role in diabetes care. In Phase 1, quantitative data revealed FCHVs unfamiliarity with the concept of diabetes, which noticeably changed in phase 2 where they were able to identify that type 2 diabetes can be caused by an unhealthy diet, inadequate exercise, and obesity (Table 2). Additionally, most FCHVs in phase 2 understood that diabetes can result from a lack of insulin or poor insulin response. In phase 1, there was less awareness of diabetes-related complications acknowledging that diabetes can cause heart attacks, heart failure, strokes, and kidney damage that improved significantly on Phase 2, although gap still persisted.

Qualitative findings from Phase 1 and Phase 2 (S3 Text) illustrated a noticeable change in FCHVs' understanding of diabetes. In Phase 1, FCHVs generally expressed basic awareness of diabetes and its management. While they had heard of diabetes, their responses indicated minimal understanding of its underlying causes, clinical manifestations, and potential health outcomes. As one participant cautiously shared,

Quote 1: "*I don't know much, but I do know that eating sweets isn't the only cause of diabetes.*" – (Ward No. 7, Phase 1)

This response hinted that although FCHVs did not have a comprehensive understanding, they knew that diabetes is multifactorial.

By Phase 2, however, participants expressed a more nuanced and accurate understanding of the condition. FCHVs were not only able to define diabetes but also described its symptoms and causes with notable clarity and confidence. One participant articulated,

**Table 1. Sociodemographic characteristics of participating FCHVs (N = 28).**

| Variables | Category | N (%) |
|---|---|---|
| Age (years) | Mean = 48, SD = ±6.79, Min = 30, Max = 59 | |
| | ≤40 | 6 (21.4) |
| | >40 | 22 (79.6) |
| Educational status | Literate | 27 (96.4) |
| | Illiterate | 1 (3.6) |
| Highest education completed | Informal Education | 2 (7.1) |
| | Primary Education (Grade 1–5) | 17 (60.7) |
| | Secondary Education (Grade 6–8) | 9 (32.1) |
| Other Occupation (Besides FCHV) | Yes | 2 (7.1) |
| | No | 26 (92.9) |
| Years of engagement as FCHVs | ≤10 years | 7 (25) |
| | >10 years | 21 (75) |

FCHV = Female Community Health Volunteer, N= group size, SD= standard deviation.

PLOS Global Public Health

**Table 2. Quantitative evaluation of FCHVs' responses.**

| Themes | Phase 1 | | | Phase 2 | | |
|---|---|---|---|---|---|---|
| | Survey (S) 1 | Survey (S) 2 | p- value | Survey (S) 3 | Survey (S) 4 | p-value |
| **Theme 1: Diabetes Knowledge** *(N [%] selecting "True" response)* | | | | | | |
| Heard about diabetes | 26 (92.9) | NA | | 28 (100) | NA | |
| Diabetes occur either from lack of insulin or when our body doesn't respond well to insulin. | 12 (42.9) | 27 (96.4) | <0.001 | 18 (64.3) | 27 (96.4) | 0.005 |
| Diabetes can cause: | | | | | | |
| Heart attack | 6 (21.4) | 23 (82.1) | <0.001 | 8 (28.6) | 7 (25.0) | 1 |
| Heart failure | 4 (14.3) | 20 (71.4) | <0.001 | 3 (10.7) | 2 (7.1) | 1 |
| Stroke | 2 (7.1) | 18 (64.3) | <0.001 | 2 (7.1) | 0 (0.0) | 0.491 |
| Damage kidney | 5 (17.9) | 17 (60.7) | 0.002 | 4 (14.3) | 3 (10.7) | 1 |
| Diabetes causes all of the above | 7 (25.0) | 17 (60.7) | 0.014 | 12 (42.9) | 20 (71.4) | 0.058 |
| Diabetes type 2 occurs from unhealthy diet, inadequate exercise and obesity. It can be cured. | 16 (57.1) | 28 (100.0) | <0.001 | 19 (67.9) | 26 (92.9) | 0.040 |
| Frequent urination and thirst are signs of diabetes. | 24 (85.7) | 28 (100.0) | 0.001 | 25 (89.3) | 28 (100.0) | 0.236 |
| **Theme 2: FCHVs confidence** *(N [%] selecting "True" response)* | | | | | | |
| FCHV has a role in diabetes management | 20 (71.4) | 28 (100.0) | 0.004 | 24 (85.7) | 24 (85.7) | 0.1118 |
| **Theme 3: Diet and Lifestyle counseling** | | | | | | |
| Skill level in counseling people to modify their lifestyle *(N [%] selecting "I can independently counsel "or "I can counsel, but only with some assistance" or both.* | 15 (53.6) | 24 (85.7) | 0.019 | 20 (71.4) | 27 (96.4) | 0.025 |
| Routinely counsel people with diabetes to modify their lifestyle *(N [%] selecting "Yes" response)* | 24 (85.7) | NA | | 12 (42.9) | NA | |
| **Theme 4: Training and education as a barrier** *(N [%] selecting "True" response)* | | | | | | |
| Lack of training and education | 24 (71.4) | NA | | 12 (42.9) | NA | |
| **Theme 5: Barriers and motivators for community-based self-management support** *(N [%] selecting "True" response)* | | | | | | |
| Community belief and attitude towards FCHVs skills | 9 (32.1) | NA | | 12 (42.9) | NA | |
| Multiple engagement of FCHVs in different health programs | 9 (32.1) | NA | | 11 (39.3) | NA | |
| Incentive and motivation | 7(25.0) | NA | | 17 (60.7) | NA | |

NA- not applicable as question was not asked.

> Quote 2: "*Diabetes is a type of non-communicable disease that leads to increased blood sugar levels. We learned this from the Health Foundation. It occurs when the sugar level in the body becomes too high. Its symptoms include unexplained weight loss, excessive thirst, dry mouth, and a tingling sensation in the hands and feet.*" – (Ward No. 6, Phase 2)

While integrating qualitative and quantitative data, it became evident that FCHVs showed progress in their understanding of diabetes, moving from general to more specific expression, suggesting that the training sessions were effective.

### Theme 2: Diet and lifestyle counseling

From the quantitative findings in Phase 1, 15 FCHVs reported that they could either independently counsel individuals about diabetes or do so with some assistance. By phase 2, the number has increased to 27, suggesting a notable improvement in confidence and skills in providing lifestyle and diabetes related counseling (Table 2).

In qualitative analysis, we noted that FCHVs at phase 1 had limited skills and confidence in diet and lifestyle counseling(S3 Text). They reported that their understanding rarely extended beyond warning against sugar consumption. They acknowledged a lack of deeper knowledge about dietary or lifestyle modifications as one participant articulated:

Quote 3: "*I only know that people with diabetes shouldn't eat sweet things. I haven't given any advice or suggestions before.*" (Ward No. 3, Phase 1)

In Phase 2 FCHVs were more aware of the broader impact of diet and lifestyle counseling on diabetes management (S3 Text). They understood the importance of self-management counseling and actively incorporated knowledge from the training into counseling community members about the significance of diet and exercise in managing blood sugar levels. One participant gave a detailed description of tailored dietary routine and regular exercise to maintain a healthy lifestyle, demonstrating a comprehensive understanding of diabetes management.

Quote 4: "*Those who have diabetes should eat their meals on time. In the morning, they should have two flat breads with vegetables on time. They should avoid rice and sweet foods. If possible, they should stick to flatbread and make sure to exercise. Otherwise, it can lead to problems later on. I share this advice with them. These days, many people have diabetes and are already aware of some of these things.*" (Ward No. 6, Phase 2)

The integration of quantitative and qualitative findings indicated that knowledge acquisition through structured training and hands on learning experience enhanced FCHVs' counseling practices.

**Theme 3: FCHVs' confidence**

FCHVs' confidence in their role to provide diabetes self-management support grew notably after phase 1 training. By phase 2, their confidence stayed high suggesting they retained that sense of ability over time. Qualitative findings showed a notable shift in FCHVs' confidence as community health educators(S3 Text). In Phase 1, FCHVs described a limited but genuine effort to share whatever knowledge they had, despite lacking formal training or comprehensive understanding.

Quote 5: "*We haven't read about this, but we share whatever little we do know with others.*" (Ward No.3, Phase 1)

In phase 2, FCHVs not only reported increased knowledge but also described feeling more capable in delivering health information. One participant noted how the training had equipped them to speak more effectively about diabetes management and counseling strategies (S3 Text).

Quote 6: "*Previously, we didn't understand much. But after receiving training, we can now counsel others more easily. We explain that diabetes is a condition that requires checkups and proper control of food intake. We're now able to share this information.*" (Ward No. 4, Phase 2)

Another participant used her lived experience with diabetes to connect with the community members and provide empathetic, informed support (S3 Text).

Quote 7: "*I have diabetes myself, so I didn't face many barriers. Since I experienced it firsthand, I know many of the symptoms. I often ask others what symptoms they are having. They tell me they feel hungrier, can't sleep well, have headaches, and they come to me for advice. When they do, I explain what diabetes is. I tell them they might feel tingling in their hands and feet, or have a dry mouth and dry nose. So, I counsel them to go get a checkup.*" (Ward No.6, Phase 2)

Mixed-methods integration suggests that FCHV's increased confidence in delivering diabetes self-management support was not only from gaining knowledge but also from their identity as a health educator in the community. The training fueled their validation of lived experience and formalized their role, leading to greater trust and impact in the community.

## Theme 4: Training and education

In phase 1, many FCHVs identified lack of training and education as a key barrier to providing self-management support within their communities. By phase 2, FCHVs were more confident and capable in their roles which suggest that the training not only improved their diabetes related knowledge but also strengthened their ability to implement that knowledge in the real-world setting.

In qualitative findings, participants reflected on their past experiences, noting a lack of structured or follow-up training and expressing the need for a more committed and systematic approach to skill-building (S3 Text). One participant shared:

> Quote 8: "*Some organizations come and provide training just once, then never return. As a result, we do not get updated.*" (Ward No. 6, Phase 1)

This sentiment was reflected in several participants' responses where they pointed a gap between initial and ongoing support to build FCHVs' capacity. One participant emphasized that without regular education, FCHVs are left unprepared to support others, underlining the critical link between training and competency(S3 Text).

> Quote 9: "*We can become capable if we receive proper training. Without knowledge, how can we offer suggestions and advice to others?*" (Ward No. 3, Phase 1)

By Phase 2, FCHVs expressed deep appreciation for the training. Many found it helpful and shared how they felt empowered to apply their knowledge. The training not only improved their technical understanding but also sparked curiosity and a stronger sense of responsibility toward implementing the learnings within their communities. One participant articulated the importance of comprehensive training as a foundation for confidence and outreach(S3 Text).

> Quote 10: "*We can only spread knowledge in the community if we receive complete training. First, we need to feel fully confident and say to ourselves, 'Yes, I can go and work in the community.*" (Ward No. 3, Phase 2)

The integration of quantitative and qualitative findings indicates that training served not only to address immediate knowledge deficits but also to cultivate a deeper commitment among FCHVs in their role as community health advocates. Their growing call for ongoing education highlights a shift from viewing training as a one-time intervention to recognizing it as a necessary foundation for sustained capacity and credibility in their community-facing roles.

## Theme 5: Barriers and motivators for community-based self-management support

In phase 1, some FCHVs identified community's belief and attitude, engagement in multiple programs and lack of incentive and motivation as barriers to provide self-management support. The barriers persisted in Phase 2 and perception of inadequate incentives worsened (Table 2).

Qualitative analysis revealed three subthemes under the main theme barriers and motivators for community-based self-management support: Workload, inadequate incentives and community's receptiveness towards counseling (S3 Text).

**Subtheme 1: Workload.** In Phase 1, FCHVs highlighted the sheer volume of responsibilities they managed, which was difficult to manage. Many described being pulled in multiple projects—supporting maternal health, child immunization, elder care, education outreach, which made it difficult to retain new information or dedicate focused time to lifestyle counseling. One participant shared how juggling dozens of tasks diluted their capacity to absorb and apply training content:

> Quote 11: "*There is a need for regular training. A one-time training is not enough. It's difficult to retain all the information from a single session over the years. We need training every six months. Since we are involved in so many different tasks, we tend to forget. As Female Community Health Volunteers, we are responsible for around 35 different types of work—ranging from caring for pregnant women to supporting the elderly.*" (Ward no. 6, Phase 1)

Although this subtheme of overwhelming work demand was prominent in Phase 1 discussions, it notably did not emerge in Phase 2, suggesting that either the training or the study's focus may have shifted FCHVs' perceptions of workload.

**Subtheme 2: Inadequate incentives.** Nearly all FCHVs in Phase 1 raised concerns about the lack of adequate incentives tied to their volunteer work. While they were deeply committed to serving their communities, they felt that modest financial or material support would bolster their motivation and help sustain their engagement over time.

Quote 12*: "We need training, we have to work day and night, so we should have services and facilities accordingly, and there should be remuneration." (Ward no. 3, Phase 1)*

Quote 13: "*There should be provision of training along with increased incentives though there is no salary for us. Rs 400 (~$3) is not sufficient for us, at least there should be provision of Rs. 1000 (~$7.10) so we will be encouraged to do our responsibility*." (Ward No.6, Phase 1)

**Subtheme 3: Community's receptiveness towards counseling.** Both Phase 1 and Phase 2 discussions shed light on how community attitudes shaped FCHVs' ability to counsel effectively, but the tenor shifted markedly over time.

In phase 1, FCHVs expressed frustration about lack of authority figures like physicians in the field with them, which could lead to community members being unreceptive to FCHVs counseling expertise, making it challenging for the FCHVs to establish their credibility and deliver impactful counseling.

Quote 14: "*Community people said that we only do counseling without providing anything. So, it would be better if there is provision of medicine*." (Ward No. 6, Phase 1)

By Phase 2, however, FCHVs described a far more receptive audience. They observed that residents were not only listening to their recommendations but also implementing them. One FCHV in Phase 2 highlighted the tangible feedback they received—community members affirming that "*the things we were saying was good*"—which reinforced the value of their counseling.

Quote 15: "*Some say that they are doing checkup as per our guidance, some said they are fine, some said their sugar level is increasing, and doctor is suggesting having control on diet. They said that the things we were saying was good*." (Ward No. 4, Phase 2)

Another participant articulated the growing receptiveness of community members towards FCHVs' self-management support:

Quote 16: "*They follow our suggestions and advice, they listen to us, they said we are right and they do checkup*." (Ward No 4, Phase 2)

The integration of qualitative and quantitative findings reveals that while systemic barriers such as workload and inadequate incentives persisted, the increasing receptiveness of the community emerged as a powerful motivator that helped sustain FCHVs' engagement in self-management support.

## Discussion

This mixed method study evaluated FCHV's knowledge, perceived self-efficacy, and barriers to providing diabetes self-management support in rural Western Nepal before and after structured training program. Guided by the HBM, we found that even a brief, skills-based training enhanced FCHVs' understanding of diabetes and its risk factors and increased their confidence in supporting community members. Although persistent barriers such as limited incentives

were identified, the training also contributed to community receptiveness towards FCHVs as diabetes-related supporters. Our findings suggest that FCHVs are both willing and able to expand their role to include diabetes care, and that short, targeted training sessions coupled with ongoing refresher training and appropriate incentives represent a feasible strategy in NCDs management in resource-constrained settings, especially if our findings are further substantiated.

A growing body of evidence supports community-based management of NCDs, especially in LMIC where there is a critical shortage of human resources [6,14,16,17,23]. A study showed that CHWs can partially fill the void from severe human resources shortage as they bridge between community and healthcare system [24]. Their role in behavior change interventions has demonstrated clinical effectiveness in multiple NCDs [6,14,16]. A randomized trial in Nepal showed significant improvements in glycemic control among individuals with diabetes who received CHW-led interventions [6]. Similarly, trials conducted in other LMICs, such as India and South Africa, have also reported that CHW-delivered programs led to improved glucose levels, blood pressure control, and adherence to treatment regimens in individuals with diabetes and other NCDs [23,25,26]. These findings highlight the potential of CHWs to not only provide self-management support but also drive measurable clinical outcomes in resource constrained settings. In this context, evaluating the readiness of CHW to deliver the service is important and timely, especially given the heterogeneity of their skills and background. The current study contributes that even a short and targeted program can significantly boost CHWs' knowledge and confidence, especially if coupled with repeat training.

FCHVs are integral members of their communities, providing various services that extend beyond a single health condition. They live in the same communities they serve and sometimes their service extends to neighboring areas. Despite establishing themselves as respected members of the community, FCHVs face challenges in a patriarchal society [13]. Our qualitative analysis highlighted that after the training, FCHVs perceived the community to be more receptive towards their counseling. This further motivated FCHVs to continue learning and providing care in their communities.

In our study, we noted barriers such as limited expertise, inadequate incentives, and heavy workloads persisted. FCHVs themselves emphasized the need for more comprehensive and frequent training to enhance their skills and credibility. Similarly, FCHVs said that the incentives as per local norms were not adequate. These barriers mirror findings from other community health worker programs including Nepal and similar settings, where lack of incentives and structural support have been shown to limit program effectiveness [27,28]. Addressing these constraints through system-level strategies such as appropriate incentive program, recognition within the health system, and provision of job benefits or counseling materials, will be essential for maintaining motivation and ensuring sustainability. At the same time, increasing receptivity of community members toward FCHVs underscores their unique potential to deliver diabetes self-management effectively if adequate support mechanisms are in place.

The findings from this study have important implications for policy and practice in Nepal and other resource-limited settings, if our findings are further substantiated. Integrating diabetes education and counseling into the existing scope of FCHVs can be feasible if training, incentives and workload and other systemic support are tailored to FCHVs needs and capacity. If substantiated, structured and targeted training modules, such as the one evaluated in this study, could be incorporated into the national FCHV training curriculum and reinforced through periodic refresher sessions.

This study has several strengths. We conducted training and evaluations in two phases. The second phase was after FCHVs had an opportunity to use the skills they learned at Phase 1. By employing a concurrent qualitative dominant embedded study design, we were able to gain depth and context through qualitative insights, while also incorporating quantitative data. We achieved thematic saturation during qualitative analysis. At the same time, the study has several limitations. First, COVID restrictions affected our recruitment, and thus, the sampling was not truly purposeful, which is preferred in qualitative work. However, participating FCHVs had a range of age, education and experience that still allowed us to rigorously evaluate our research questions. Furthermore, our evaluation was limited to one district. We encourage additional studies to evaluate transferability of our findings to other places in Nepal, especially given the potential heterogeneity of FCHVs knowledge, skills and education. Second, the training primarily emphasized

knowledge rather than skills building, with limited opportunity for structured assessments of counseling fidelity, largely due to resource constraints. Although concepts such as motivational interviewing and shared decision-making were introduced, the training was not designed to provide a deeper understanding or supervised practice of these methods. Third, the COVID-19 pandemic created challenges for community engagement, contributing to variability in FCHVs' opportunities to apply their training between phases. Fourth, there were barriers persistent in phase 2, which were not addressed by the scope of the training. Fifth, the quantitative evaluation was underpowered for inferential testing, so the results should be interpreted as descriptive trends rather than definitive effects that added to the detailed qualitative evaluation. Furthermore, although we used a rigorous process to develop the survey, we did not conduct formal psychometric evaluation. Finally, as with many self-report surveys, there may have been risks of social desirability bias. Despite these limitations, this study provides important insights into the potential for brief, repetitive, skills-oriented trainings for FCHVs in rural Nepal.

## Conclusions

Our findings highlighted the potential of focused and repetitive training in FCHVs' role in the prevention and management of diabetes in resource limited settings. However, sustainability and impact of FCHVs led interventions are shaped by the structural and social conditions such as incentives, workload and the receptiveness of the community in FCHVs' capacity in diabetes care. Transferability of these findings to other areas of Nepal should be evaluated. As burden of diabetes and other NCDs continue to rise, strengthening interventions led by FCHVs that are socially and structurally supported can offer a way forward for more sustainable and equitable care.

## Supporting information

**S1 Text. Quantitative Survey Questionnaire.**
(DOCX)

**S2 Text. Qualitative Interview Guide.**
(DOCX)

**S3 Text. Theme-subthemes and Quotes in English.**
(XLSX)

**S4 Text. Translated Quotes (with Original Nepali Text).**
(DOCX)

## Author contributions

**Conceptualization:** Ada Thapa, Susagya Bhusal, Sudim Sharma, Yashashwi Pokharel.

**Data curation:** Susagya Bhusal, Adipti Pantha, Sudim Sharma.

**Formal analysis:** Ada Thapa, Susagya Bhusal, Austin Seals.

**Methodology:** Ada Thapa, Susagya Bhusal, Yashashwi Pokharel.

**Project administration:** Susagya Bhusal, Adipti Pantha.

**Supervision:** Yashashwi Pokharel.

**Writing – original draft:** Ada Thapa, Susagya Bhusal, Yashashwi Pokharel.

**Writing – review & editing:** Ada Thapa, Susagya Bhusal, Omkar Jha, Sudim Sharma, Austin Seals, Madhav K. C., Archana Shrestha, Bibhav Acharya, Yashashwi Pokharel.

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
