## [Decision Letter · Decision Letter 0]

25 Dec 2025

PGPH-D-25-03684

Female community health volunteers’ knowledge and confidence in providing community-based diabetes self-management support in Nepal: A biphasic mixed method evaluation

Dear Dr. Bhusal,

Thank you for submitting your manuscript to PLOS Global Public Health. After careful consideration, we feel that it has merit but does not fully meet PLOS Global Public Health’s publication criteria as it currently stands. Therefore, we invite you to submit a revised version of the manuscript that addresses the points raised during the review process.

We look forward to receiving your revised manuscript.

Kind regards,

Baldeep Kaur Dhaliwal, PhD

Academic Editor

Journal Requirements:

i. Please clarify all sources of financial support for your study. List the grants, grant numbers, and organizations that funded your study, including funding received from your institution. Please note that suppliers of material support, including research materials, should be recognized in the Acknowledgements section rather than in the Financial Disclosure.

ii. State the initials, alongside each funding source, of each author to receive each grant. For example: "This work was supported by the National Institutes of Health (####### to AM; ###### to CJ) and the National Science Foundation (###### to AM)."

iii. State what role the funders took in the study. If the funders had no role in your study, please state: “The funders had no role in study design, data collection and analysis, decision to publish, or preparation of the manuscript.”

iv. If any authors received a salary from any of your funders, please state which authors and which funders.

2. Please send a completed 'Competing Interests' statement, including any COIs declared by your co-authors. If you have no competing interests to declare, please state "The authors have declared that no competing interests exist". Otherwise please declare all competing interests beginning with the statement "I have read the journal's policy and the authors of this manuscript have the following competing interests:"

3. Please ensure that your Ethics Statement is available in its entirety at the beginning of your Methods section, under a subheading 'Ethics Statement'.

4. In the online submission form, you indicated that “Data are available upon reasonable request.”

3. Uploaded as supplementary information.

Reviewers' comments:

Reviewer's Responses to Questions

**Comments to the Author**

1. Does this manuscript meet PLOS Global Public Health’s publication criteria?

Reviewer #1: Yes

Reviewer #2: Yes

2. Has the statistical analysis been performed appropriately and rigorously?

Reviewer #1: Yes

Reviewer #2: Yes

3. Have the authors made all data underlying the findings in their manuscript fully available (please refer to the Data Availability Statement at the start of the manuscript PDF file)?

Reviewer #1: Yes

Reviewer #2: Yes

4. Is the manuscript presented in an intelligible fashion and written in standard English?

Reviewer #1: Yes

Reviewer #2: Yes

Reviewer #1: The manuscript explores an important topic. It shows how short trainings can help Female Community Health Volunteers (FCHVs) support diabetes care in rural Nepal. The biphasic mixed-methods design and Health Belief Model are suitable. Results suggest gains in knowledge and confidence. These findings are useful for low-resource settings. However, the study is small and affected by COVID-19. It works best as a pilot study. Major revisions are needed for clarity and caution.

Strengths: The topic is timely and practical. It builds on Nepal’s existing CHW system for NCD care. The 11-month follow-up and refresher training add value. Mixed-methods integration is done well. Participant details and themes are clear. Recruitment and ethics are transparently reported.

Major Concerns:

1) The sample is small (n=28 from one district, four wards). Selection was convenience-based due to COVID-19 rules. This limits representativeness. Broad claims about FCHVs in Nepal need toning down. Frame the work clearly as exploratory or pilot.

2) The study occurred during COVID-19 restrictions. These affected recruitment, group size, and possibly outcomes. The current brief mention is not enough. Expand the limitations section with specific pandemic effects.

3) No control group exists. Phase 2 results combine initial training, field experience, and refresher effects. This makes it hard to isolate the impact of one short training.

Recommendations:

Present the study as pilot or exploratory. Avoid wide generalizations. Expand limitations to cover COVID-19 impact, selection bias, lack of control and mixed effects from refresher. Balance conclusions by noting persistent barriers.

This is a helpful pilot with clear potential. More transparency and caution will make it suitable for publication. I look forward to the revised version.

Reviewer #2: Congratulations on the success of your intervention—this is a strong and well-written article.

I have a few minor questions and suggestions that may be helpful to address prior to publication:

1. You note an 11-month interval between Phases I and II in both the abstract (line 44) and the Methods section (line 157); however, the period from January to October appears to be nine months. Please clarify this discrepancy in the timeline. Additionally, the mention of date in the abstract may not be necessary. If you decide to keep it, please also include the dates for Phase II.

2. The introduction effectively frames diabetes as a global concern and highlights the role of community health workers as a bridge between communities and the health system. It would strengthen the paper to include additional context on the burden of diabetes in Nepal—particularly in the study region—and the unique causes and risk factors relevant to this setting.

3. While the study context is described, the specific purpose or objective of the paper is not stated explicitly. Clarifying this in the introduction would help orient the reader.

4. I am interested in the 9-item survey assessing diabetes knowledge and confidence (line 140). Was this a newly developed instrument or an adaptation of an existing, validated diabetes knowledge questionnaire? If self-developed, could you briefly describe the steps taken to assess its validity?

5. The study design is described as a “concurrent, embedded mixed-methods” approach in the abstract (line 41) and Methods section (line 186), but as “concurrent” in the introduction (line 93). Please ensure consistency in terminology throughout the manuscript.

6. The formatting of the brackets in the final column of Table 1 appears inconsistent (this may be a display issue on my end). Please review this and consider standardizing the table design for consistency.

I hope these comments are helpful, and I look forward to seeing the revised version.

**Do you want your identity to be public for this peer review?** For information about this choice, including consent withdrawal, please see our Privacy Policy

Reviewer #1: No

Reviewer #2: No

---

## [Decision Letter · Decision Letter 1]

18 Feb 2026

Female community health volunteers’ knowledge and confidence in providing community-based diabetes self-management support in Nepal: A biphasic mixed method evaluation

PGPH-D-25-03684R1

Dear Bhusal,

We are pleased to inform you that your manuscript 'Female community health volunteers’ knowledge and confidence in providing community-based diabetes self-management support in Nepal: A biphasic mixed method evaluation' has been provisionally accepted for publication in PLOS Global Public Health.

Best regards,

Baldeep Kaur Dhaliwal, PhD

Academic Editor

Reviewer Comments (if any, and for reference):

Reviewer's Responses to Questions

**Comments to the Author**

Reviewer #1: All comments have been addressed

publication criteria?

Reviewer #1: Yes

3. Has the statistical analysis been performed appropriately and rigorously?

Reviewer #1: Yes

4. Have the authors made all data underlying the findings in their manuscript fully available (please refer to the Data Availability Statement at the start of the manuscript PDF file)?

Reviewer #1: Yes

5. Is the manuscript presented in an intelligible fashion and written in standard English?

Reviewer #1: Yes

Reviewer #1: The authors have provided appropriate references supporting their decision not to classify this study as a pilot or exploratory work. I am satisfied with the evidence and resources they have presented. The remaining comments have also been thoroughly addressed with strong supporting references. All concerns have been resolved.

Best wishes!

**Do you want your identity to be public for this peer review?** For information about this choice, including consent withdrawal, please see our Privacy Policy

Reviewer #1: **Yes:** Amrit Gaire
